# “Step by Step We Were Okay Now”: An Exploration of the Impact of Social Connectedness on the Well-Being of Congolese and Iraqi Refugee Women Resettled in the United States

**DOI:** 10.3390/ijerph20075324

**Published:** 2023-03-30

**Authors:** Caitlin Bletscher, Sara Spiers

**Affiliations:** Department of Human Development, Washington State University, Vancouver, WA 98686, USA

**Keywords:** gender, refugee, resettlement, social capital, belonging, isolation

## Abstract

Little is known about the gendered impacts of the displacement and resettlement process. Women are known to struggle more with feelings of belonging and the creation of social networks to access essential information, resources, and social and emotional support to enhance their overall health and well-being. The purpose of the present study was to qualitatively explore female refugee perceptions of belongingness and social connectedness post-resettlement into their U.S. host community. Conducted between January and June of 2016, through the partnership of multiple governmental, nonprofit, and community-based organizations, two female focus groups were conducted among Congolese (*n* = 6) and Iraqi (*n* = 6) U.S. resettled refugees. Descriptive surveys were distributed to participants, providing valuable insights into participant demographics and indicators that could impact the integration process (i.e., age, language, country of origin, ethnicity, education, length of time spent in the U.S., housing). Participants discussed the social connections (individuals, communities, organizations) that acted as facilitators or inhibitors of developing social capital. The importance of building strong transformational bonding (family members, other refugees) and bridging (host community) relationships, alongside transactional ties with linking agencies (resettlement social services), was critical for a positive resettlement experience. The strength of their network ties among these social connections contributed to their perceptions of belongingness and well-being post-resettlement into their host community.

## 1. Introduction

The well-being of an increasing number of displaced people has become a global public health concern [1,2]. According to the Guiding Principles on Internal Displacement [3], displaced people include:

“Persons or groups of persons who have been forced or obliged to flee or to leave their homes or places of habitual residence, in particular as a result of or in order to avoid the effects of armed conflict, situations of generalized violence, violations of human rights or natural or human-made disasters, and who have not crossed an internationally recognized border.”

Such populations experience loss of family, social support, and identity. Most displaced people are resettled to a secondary country, while about 1% of the migrant population is resettled into a third host country such as the United States (U.S.). This compounding displacement creates additional complications and barriers to the resettlement process and integration post-resettlement.

Gender has been identified as one such barrier to successful resettlement [4]. Even though gender has been consistently identified as an individual indicator to successful integration among migrant populations [5,6], most research and discourse on migration instead tells a homogenous gendered story [7,8]. This narrative shows refugees as unitary, non-gendered entities, assuming a ‘gender blind’ stance that ignores the ways that gender shapes the experiences and risks in migration [9].

Thus, a shift of attention or a ‘womanist framework’ [10] is needed to understand the experiences of migrant women [9], due to the millions of women experiencing forced displacement worldwide [4]. Historically, about 50% of the world refugee population are women [11], and numbers continue to increase. Female refugees face significant obstacles to coping with the many challenges of migration in comparison to their male counterparts. Obstacles include fear and persecution [4], gender-based discrimination and physical and/or sexual violence [5,11,12,13,14,15,16], cultural displacement, inequitable gender norms or confluence of cultural gender role expectations, language barriers, lack of resources and education, and an increase of structural and situational stressors [1,5,7,16,17]. Women’s social identities also reside in disadvantaged locations in the hierarchical social sphere [18,19], creating significant social tensions [17].

Upon resettling to their third host communities, refugee women face a wide array of challenges and obstacles, especially for refugees from non-western and non–white countries [20]. After female refugees migrate, they experience conflict within their personal identify development. On one hand, they have their perceptions of their ethnic, religious, and linguistic identities; on the other hand, they are embedded within their new host community [4], which has different expectations, which can significantly impact their sense of well-being [20].

Because women often migrate as dependents of their male partner or family members, the focus tends to fall on the male migrants’ needs, dismissing women’s unique health concerns and well-being [21], not to mention the significant challenges facing resettled women with navigating the U.S. healthcare system [22,23]. Thus, during resettlement, female refugees face significant barriers to appropriate healthcare, due to a lack of health literacy, cultural differences, communication issues, and challenges with access [24]. This highlights why gender (females) has been significantly positively associated with symptoms of depression, anxiety, Post-Traumatic Stress Disorder, and risky behavioral problems such as Alcohol Use Disorder [25,26].

Many times, refugee women are caught in a ‘cycle of isolation’ upon resettlement, stemming from religious and ethnic backgrounds, alongside the host country’s social, political, and institutional processes [27]. This cycle is continuously reinforced throughout the integration process, through ongoing internal ethnic factors and external marginalizing factors [27] such as discrimination, racism, and prejudice from their host community.

While the research on what drives public perceptions and attitudes toward refugee populations (versus immigrants at large) is limited [28], findings have showcased negative perceptions such as discrimination and profiling towards resettled refugees in the U.S. [29,30], mainly as potential security threats to U.S. safety and well-being [28]. Results of a 2016 study of American adults showed preferences towards Syrian refugees who are female, highly skilled, English speaking, and Christian [28]. Religious and gender identities were significant variables among American preferences of their newcomers. Religious bias towards Christianity was especially significant [28], in agreement with previous research on American anti-Muslim bias and discrimination [31], especially towards women [4,32,33]. These findings build on previous literature that found that Muslim women experience more discrimination against their religious group than their Muslim male counterparts [34].

As the research suggests, the refugee migration experience significantly differs by gender, where each gender responds differently and their corresponding realities impact all stages of migration (pre-, during, and post-resettlement) [9]. Not only do refugee women face significant obstacles to successful resettlement, but the isolation and marginalization that female refugees experience as the ‘outsider’ among their new host community can further harm or inhibit access to resources to support their well-being. In order to better understand their integration process into their new community, the following migration literature review provides insights into gender inequities throughout the resettlement process (focusing on post-resettlement), with intentional focus on refugee social connectedness and perceptions of belongingness.

### 1.1. Social Connectedness and Belongingness among Refugee Populations

Kohut’s [35] self-psychology theory defined social connectedness as a construct which allows people to feel they are “being human among humans” (p. 200) in order to identify with others who may be dissimilar. A decrease in connectedness leads to isolation, where the individual finds it hard to accept his or her social role [36]. Therefore, if refugees have limited social connectedness and limited feelings of belonging among their new communities, the more isolated they may become.

An individual’s formation of social networks significantly benefits their overall well-being. Individuals with a high sense of connectedness can develop relationships easily, while others without social connectedness may experience low self-esteem, anxiety, and depression [36]. With these benefits in mind, Ager & Strang’s [6] Indicators of Integration framework demonstrated that the values of social connections shared among refugees are vital to positive integration experiences post-resettlement.

Similarly, a significant amount of research has been dedicated to exploring the impact of belongingness or ‘(non)belonging’ on overall refugee well-being (i.e., [37,38]). Belongingness is not only linked to community acceptance [6], but it is also one of the most important components of resettlement [37]. Considering the transitory nature of refugee populations, where connections are lost as individuals are displaced from their homes, they often form relations of (non)belonging [37] with new places and communities that are not always receptive to their arrival [38].

Carey-Wood et al. [39] conducted a survey among refugees resettled in the United Kingdom exploring their perceptions and attitudes of integration, concluding that seeing Britain as their permanent home was a significant contributing factor to their sense of belonging. Similarly, Spoonely et al. [40] suggested that the acceptance of identity and individuality is a dimension of feeling part of one’s host community. They illustrated the critical need for refugees to first feel a sense of connectedness with their host community before they can start developing social connections elsewhere. Without this initial sense of belongingness, refugees are more susceptible to vulnerability, unable to access essential support, resources, or information obtained from their social networks.

Yuval-Davis et al. [41] suggested that there is a complex relationship between belongingness and social cohesion among British migrants and their host community. Their research concluded that belongingness is context-specific, including demographic information like one’s country of origin, host country, religion, ethnicity, and culture. Therefore, migrants have the potential to achieve social cohesion among their new community by socially excluding migrants who do not entirely ‘integrate’ (i.e., adopt the language and culture) into the host community.

### 1.2. Female Refugees and Social Capital

Research has shown that refugee women hold successful resilient strategies to overcome the disproportionate number of obstacles they face [1,7]. Most importantly, the makeup of their social networks [11,19,42], as well as their social positioning [12] and social experiences [5], have been shown to influence coping ability and can significantly impact (support or inhibit) social, emotional [5,42], mental [5,24,43], and physical [5,11] health and well-being. Social connections have been shown to not only reduce isolation, but also increase access to healthcare [24].

Previous research has showcased the impacts of gender on bonding, bridging, and linking social capital. Zetter et al. [44] concluded that refugee women are more likely to generate *bonding social capital*, or social connections with similar populations (i.e., same resettled ethnic groups, family members). In their longitudinal photo voice study among U.S.-resettled Congolese refugee women, Saksena and McMorrow [5] revealed that social bonding within their own ethnic community (especially those living in close proximity) combatted some of the challenges with integration experienced by participants. Congolese women would turn to each not only for social support, but also for economic and physical support (i.e., meals).

The findings from Saksena and McMorrow [5] suggest that members of multi-cultural communities should also engage in social interactions among differing ethnic groups (*bridging social capital*) in order to expand their social capital; in turn, generating greater reciprocity and increased levels of trust within their host community [45]. Without these opportunities to develop social networks among differing ethnic groups, refugees may develop strong bonds of solidarity and segregation with similar groups, impeding access to resources, limiting social mobility, reducing social trust, and possibly perpetuating social exclusion [46]. Bridging social capital, through community-based interactions like religious activities or even friendly neighborhood conversations [6], provide important opportunities for belongingness, security [6], and essential emotional, social [47], mental [48], and health-related [24] support.

Female refugees have also disproportionally lacked *linking social capital,* such as career readiness or education courses [49], which provides social connectedness with hierarchical institutions of power and access to support [50]. For example, female refugees tend to have less access to social services that provide English language classes compared to their male counterparts [51] and hold a more silenced voice among service providers and governmental agencies [47], missing out on important connections that could possibly increase belongingness, overall well-being, or even opportunities for an increase in power and participation in their host community.

Kingsbury et al. [12] conducted personal social network analysis among U.S.-resettled Bhutanese recent mothers. They explored social network ties, demographic characteristics of ties, frequency of communication, length of relationships, and the strength of social networks among mothers prior to being resettled and upon giving birth in the U.S. Researchers found that three close personal connections were essential during their pregnancy, including family members that participants knew prior to resettlement. Considering public health strategies, results suggest that family is an important social connection for refugees that could be leveraged to improve the flow of information and resources, supporting female refugee well-being during and after the resettlement process. The authors’ findings support other migration research (i.e., [51]) that suggests that social connections in general are not only useful in ensuring refugees’ health and well-being, but can also be useful in developing effective programming, services, and policy surrounding migration and resettlement.

While the establishment and development of social ties after resettlement has been shown to positively impact female refugee experiences and remains critical to the integration experience [52], there is limited research addressing the social networks themselves, including their reliance on those social connections and who is and is not part of those connections [53]. While social connectedness is not the only source of support for female refugees (i.e., religion and spiritually [4,42,54]), it is clearly essential in their overall health and well-being.

### 1.3. Critical Feminist Approach

Acknowledging global and local systems (political, economic, social structures) of gender inequities and oppression, Critical Feminist Perspective (CFP) centralizes women’s unique lived experiences through an intersectional approach to gender identity [8,55,56,57]. In alignment with this approach, the exploration of immigrant and refugee women’s health and well-being requires:

“... new ways of inquiry, ones that move beyond the traditional ways of conceptualizing and operationalizing research... creat[ing] space for the exploration of how various dimensions of social identity, such as race, gender, and class, as well as education, citizenship, and geographical locations, intersect to influence the health of immigrant and refugee women” (p. 33) [13].

Intersectionality has been useful in analyzing power among relationships developed while in migration, which are the source of much social inequality and discrimination, especially for women fleeing persecution [58]. The authors therefore utilized this CFP lens to frame the present study, combatting the sentiment that genders form homogenous experiences by examining shifts in identity, power, social positions, and oppression [57,59] post-resettlement.

Intersectionality acknowledges that gender, race, class, ethnicity, sexuality, and nationality form interdependent systems of power that continuously shape individuals’ lives [55,60,61]. Theorists within this field have rejected the separation of these categories and have instead vocalized the need to both engage with and analyze how these groups intersect among one another symbolically and in practice [61,62,63]. Therefore, social service providers and policymakers alike should not only become familiar with intersectionality but must also become aware of its significance and familiar with its practicality when developing policy and implementing programming among vulnerable populations. Without looking through an intersectional lens, governmental institutions and social service agencies run the risk of further discrimination, possibly creating or enhancing inequitable and oppressive power relations among its communities (whether intentional or not).

A vast amount of research (especially among governmental frameworks) has been conducted to provide an operational standardization for ‘successful’ (positive outcomes) integration post-resettlement, where refugees progress towards fully engaged participation in their host country [64]. Although the framework for refugee integration by Ager and Strang [6] commissioned by the United Kingdom provided foundational scholarship on integration, ultimately, there is no all-encompassing meaning or definition of the term, which varies among host countries [65]. Without being clearly defined, intersectionality has been used to combat the complexity of integration within migration literature. A need has risen to look beyond simplistic language in order to more fully capture the complexity of the integration and resettlement process; “this level of complexity regarding the understanding of integration is reinforced by a lack of understanding in common discussion as to what exactly refugees are expected to integrate into and how” (p. 5) [6]. Without looking beyond some of these static categories, practitioners and policymakers run the risk of ignoring the complex histories and multitude of identities that go into community formation among female refugees [66].

The impact of complex intersections of social identities on the health of immigrant and refugee women has grown in recognition in recent decades [18]. Researchers have suggested that indicators and definitions of integration vary across historical contexts, regional locations [40], and cultural dimensions [67]. Social service providers, healthcare professionals, and policymakers must consider the cultural dimensions of social injustice and the many systemic and individual barriers to integration that migrant populations face, including forms of indifference, discrimination, and social exclusion from their host community [40].

Previous migration research has highlighted these many factors that impact the migration and integration process and overall well-being of refugee populations; addressed in this present study were categories of city or residence, language, age, English comprehension, marital status, employment, education, length of time living in the U.S., and housing. Intersectionality was specifically addressed through the analysis of both systemic (qualitatively, focus groups) and individual (demographic survey) indicators that lead to refugee integration experiences post-resettlement. This research provides an important contribution to the existing literature (i.e., [11,19,47]) that acknowledges the significance of social connectedness as a determinant of refugee health and well-being for women, contributing to their resettlement outcomes.

### 1.4. Congolese and Iraqi Female Refugees

A refugee’s ethnicity and cultural context impacts their integration outcomes post-resettlement [5,45,68]. Critically understanding the cultural and historical components of a population is the first step to facilitating more positive integration outcomes [69]. This study specifically explored Congolese and Iraqi female U.S.-resettled refugees. Around the time of this study (2014–2015), the largest groups of refugees admitted into the U.S. were from Iraq, Burma, Somalia, and Bhutan [70,71]. Acknowledging the differences in culture among the study’s participants is an essential component for understanding the meaning that is given to gender, their migration experience, and their social networks—all of which impact social inequities. Therefore, the following brief cultural background provides a foundation for this study’s specific population of Congolese and Iraqi refugees resettled in the U.S.

**Congolese Refugees.** During the time of this study, one of the United Nations’ (U.N.) largest foci in Africa was the Democratic Republic of the Congo (D.R.C.), as they tried desperately to find homes for more than 50,000 Congolese refugees over the upcoming years [72]. Despite the fact that over 470,000 Congolese people took refuge in the D.R.C.’s nine neighboring countries [73], due to the heightened unrest, many needed to be resettled outside of the continent of Africa into a third host country. Over the span of five years during the time of this study (2013–2018), the U.S. Bureau of Population, Refugees, and Migration announced that tens of thousands of displaced Congolese refugees would be resettled in the U.S. [71,72,73], joining the 10,000+ that were already resettled in the U.S. since 2001 [73].

Two (+) decades of armed conflict and unrest in the D.R.C has recently contributed factor to the increasing numbers of Congolese refugees. A majority of displaced Congolese people are ethnic minorities from the Eastern D.R.C [72]. The conflict stemmed from the 1996 Rwandan invasion in pursuit of those who had taken refuge from the genocide in the Eastern D.R.C., followed by the first and second Congolese wars in 1996 and 1998 [73]. The heightened brutality and mass numbers of various countries involved in conflict during this time procured the name ‘Africa’s World War’ [73], the deadliest conflict since World War II [73]. Despite the official end of the second Congo War in 2003, the Congolese continued to suffer the aftermath of decades of conflict and violence [74]; by 2013, more than 2.4 million Congolese were displaced [73]. Gender is essential when considering displacement and conflict in the D.R.C. due to gender-based violence; sexual violence has been and is still currently so commonly used as a weapon of war in the country that “human rights groups have called the area ‘the most dangerous place in the world to be a woman’” (p. 2) [73].

A majority of Congolese refugees come to the U.S. with work experience as farmers and herders from rural areas, unskilled workers and professionals from urban areas, and small traders [73]. Similar to other refugee ethnic groups, the Congolese have arrived in the U.S. with high expectations, without preparation for the difficulties that await them in the U.S. [73]. Compounding these difficulties is ethnicity; research has shown social and cultural discrimination is especially heightened for Black female refugees in the United States [5] and is a central component of their integration experience [20].

During the time of this study, almost all (96%) Congolese refugees were pronounced Christians, finding churches to be a tremendous resource for both fiscal and emotional support upon their arrival, similar to the function of churches in the D.R.C. [73]. Due to past history of personal violence and trauma in their country, many Congolese have shown high rates of Post-Traumatic Stress Disorder, which is often masked due to cultural values and attitudes [73]. Therefore, many U.S.-resettled Congolese turn to religious faith for healing and support as an important role in personal and community life [73].

**Iraqi Refugees.** Just prior to the time of this study, “the U.N. High Commissioner for Refugees (UNHCR) estimated that more than four million Iraqis [were] displaced by the war in Iraq and its aftermath” (p. 5) [75]. During this time, about half (two million) of these Iraqi refugees fled nearby to secondary host countries of Syria, Jordan, Egypt, Iran, Lebanon, and Turkey. However, it’s important to note that Iraqi refugees were resettled to the U.S. even prior to the Iraq war, with an estimated 12,000 admitted during the 1991 Gulf War [75].

Refugees have fled from Iraq for various reasons, including ethnic, religious, and political persecution. Ethnic targeting has included Sunni-Shi’I violence against Muslims, religious targeting of non-Muslim minorities (Baha’is, Christians, Jews, Sabean-Mandaeans, and Yazidis), and political targeting of supporters of the former regime, the insurgency, the current Iraqi government, or multinational forces [74]. Since most Iraqi communities are targets for other groups, the country has experienced numerous breaches to human rights and is not always able to protect its citizens; hence, resettlement back to Iraq or in a neighboring country is not available to most refugees. Therefore, Iraqi people must flee to a third host country resettlement, such as the U.S. [75]. Since large-scale Iraqi refugee processing for admission into the U.S. was initiated in February of 2007, the U.S. committed to increasing Iraqi acceptance numbers into the U.S. as a worldwide commitment during the time of this study [76].

## 2. Materials and Methods

### 2.1. Purpose

The purpose of the present study was to qualitatively explore female refugee perceptions of belongingness and social connectedness post-resettlement into their U.S. host community.

### 2.2. Study Design

This study was conducted between January and June of 2016, with the partnership of the U.S. Department of Children and Family Service’s Refugee Services Program, three Voluntary Agencies, one governmental agency, three community-based organizations, two nonprofit organizations, and two churches. These partnering organizations were essential due to their established trust, credibility, and ongoing contact with the study’s hard to reach population.

Research was conducted in two metropolitan cities that resettle some of the largest number of refugees in the state of Florida. Two female focus groups were conducted in this study, consisting of Iraqi (*n* = 6) and Congolese (*n* = 6) refugee participants. Depending on the participants’ English language fluency, the guidance of the focus group was either conducted partially in English or entirely translated between English and participant native language with use of community representatives. Representative leaders within the local community were identified by a governmental liaison and confirmed by local service provider partners in the study. The representatives were heavily involved in participant recruitment, translating back-and-forth during focus groups, and data analysis through member checking, ensuring the retention of the original meaning of the researcher questions and participant responses, and further explaining concepts that participants used that were unfamiliar to the researcher [77]. After concluding each workshop, the researcher debriefed with the representative to determine the accuracy of their summary. Although beneficial to have a translator as a member of the participants’ own ethnic community (i.e., trust, *bonding social capital*), the authors note the significant challenges that exist with interpreters in terms of bias, power dynamics, language diversity, and cultural approaches traditional to Western research practices [78].

Prior to the focus groups, each participant received two printed copies of the informed consent document (in their native language) and questionnaire (in their native language). These surveys provided valuable insights into participant demographics, as well as additional indicators that could impact the integration process (i.e., age, language, country of origin, ethnicity, education, length of time spent in the U.S., housing). This descriptive survey was essential in understanding how women experience multiple domains [3]. Quantitative data from the descriptive survey were analyzed with frequencies and mean scores.

Using a postpositivist perspective, the questions for the open-ended, semi-structured focus group were determined utilizing an expert-oriented approach [79]; questions were designed based on the professional expertise of the panel of experts and previous literature (i.e., [38]). The guide for the focus group instrument and procedure can be found in the entire report [80]. Each focus group consisted of 6 participants each and lasted approximately 1.5 h.

When considering the environment of working with refugee populations, the most culturally appropriate means of collecting data has been shown to be conducted through focus groups, storytelling, or outreach conversations, versus solely surveys or scalar types of evaluation [81]. By utilizing focus groups, this study allowed space for “participant ideas to build from one interviewee to another” (p. 445) [82] and allowed for the researchers to explore how people, organizations, and structures impact the refugee resettlement process [83]. Furthermore, due to the complex nature of the relationships formed throughout the integration process, qualitative data best suited the nature of this study, providing the richest contextual data necessary to understand the social connections that impact refugee well-being. This methodology allowed for ‘interpreting interpretations’ of the understanding of the relation of migration, rather than solely explaining it [83]. Qualitative methodologies allowed the researchers to obtain a “fuller expression of refugee experiences in their own terms” and were “essential in revealing subjective aspects of integration… [which was significant because] how refugees feel about their experiences was as important an indicator as are objective indicators of adaption such as employment, income, and socio-economic mobility” (p. 53) [81].

The focus groups were audio recorded and then transcribed into English utilizing an online translation service. The analysis was completed on the translated transcripts. The authors note the limitations of this process due to the many challenges facing translators and interpreters with possible misunderstandings or differences in semantics [78], with the potential of even contributing to further participant marginalization [84]. With this consideration, the authors note that migrant researchers using both qualitative and quantitative data must acknowledge that both language and culture play a significant role in successfully addressing the needs of U.S.-resettled refugees [85].

The authors applied Braun and Clarke’s [86,87] six-phase thematic analysis approach to gain familiarity with the data, code the data, and interpret the data. In order to become familiar with all data sources (phase 1 [86,87]), the authors independently read and re-read the focus group transcripts thoroughly and reread the focus group guides before beginning the qualitative analysis [86,87]. This step enhances the study’s validity, ensuring that the findings of the study accurately reflect the data [86,87,88]. The authors then followed phase 2 and independently identified codes within the focus group data [86,87]. Throughout the interpretation process, the researchers sought to be empathetic in order to consider how the world looks to refugee populations [89], as well as reflexivity, considering the power of the research findings on the population and limitations of the authors’ perspectives [90]. To ensure rigor in the analysis process, the authors met to discuss their codes through an iterative comparison process, which allowed them to find themes (phase 3) that were specific to each focus group, as well as emerging themes that cut across all focus groups [86,87]. Themes were revisited and refined once all codes were identified.

### 2.3. Sample

Refugee participants included Congolese and Iraqi refugees currently residing in the state of Florida in the U.S. (*n* = 12), who were recruited from local partnering organizations. Informed consent was obtained from all subjects involved in the study. Demographics among Congolese and Iraqi participants varied, especially among self-reported city of residence, age, employment status, education, and number of individuals living in their household. For more detailed information regarding a comparison of participant demographics among the two focus groups, please see Table 1.

It is important to note that a majority of women in this study were unemployed and lived in semi-permanent housing (rented). In addition to housing concerns, their lived experiences have the potential to place them in economic dependence and restrict mobility with members of their family or community, increasing their overall vulnerability [4]. Additionally, considering the higher amount of educational experience among Congolese women in the present study, previous research has found that, despite arriving with higher education and professional credentials, some refugee women may not get such credentials recognized in the U.S. [31]. Similarly, Stempel and Alemi [31] found that less educated women, as well as women whose received education outside of the U.S., were most targeted or harmed by discrimination among multiple refugee groups.

### 2.4. United States Political Climate

It is important to note that the present study was conducted during a very polarizing time in public discourse on migration, with the 2016 U.S. presidential election. However, considering the years that have passed since the time of this study, ongoing shifts in the U.S. political climate surrounding migration continue to exist. Thus, the results of this study provide service providers and policymakers with a necessary foundation for existing (and in many cases, *growing*) vulnerabilities among U.S.-resettled refugees. For example, the drastic decrease in U.S. refugee resettlement (and migration at large) during the global pandemic, as well as the harsh cuts to the refugee resettlement program (including significant resettlement agency downsizing and closures) by the Trump administration developed further vulnerabilities for the U.S. Refugee Program, undermining the U.S. placement as the world’s top country for third-country refugee admissions [91].

Lastly, considering the ongoing (if not heightened) opposing discourse on immigration across partisan lines, this study holds significant weight into the perceived experiences of female refugees. Even as of recent, there continues to be an increase in stigmatized targeting of Muslim religious identities as migration becomes more politicized [31]. Considering current U.S. President Biden’s hope to expand the domestic resettlement program [91], the results of this study provide insights into successful strategies and policies that could combat current inequities among female refugees resettled or in the process of resettling in the U.S.

## 3. Results

Despite the plethora of evidence outlining the adversities faced by female refugees post-resettlement, both focus groups mentioned the welcoming nature they experienced in their community (especially Iraqi participants). Although this connectedness started initially upon resettlement, developing a sense of belongingness required a significant amount of time and trust, especially among their host community and service providers. Established and reciprocal trust was essential for the development of bonding, bridging, and linking social capital.

The development of *bonding* social capital varied among focus groups. Although both focus groups found essential social connections through church and spiritual outlets, important bonding networks for Congolese participants were established at work and developed among fellow female refugees, while Iraqi networks were strong among family members and other refugee groups (without gender specificity). Differences in language and culture made it challenging for both focus groups to develop strong (*bridging*) ties with their host community. Despite these barriers, Iraqi women still vocalized important social connections among American community members to provide them with practical support and assistance.

Lastly, while Iraqi participants gleaned essential information and resources through their *linking* networks, Congolese participants emphasized their insufficiencies in cultural awareness, communication, and genuine care/concern; similar to *bridging* social capital, lack of English language fluency was also a barrier to support from social connections. Both focus groups emphasized the lack of quality resources provided by social service agencies, resulting in increased isolation. This lack of *linking* social capital created self-dependency and further perpetuated their isolation, creating additional stressors for participants (i.e., employment, housing, physical healthy) that negatively impacted their well-being post-resettlement, especially among Congolese participants.

In the discussion that follows, the authors further elaborate on these emerging themes, drawing from participant voices to highlight perceptions of belongingness and reciprocal trust among their social connections. Table 2 summarizes the themes and codes that emerged from both focus groups.

### 3.1. Belongingness and Trust

Despite the many adversities faced by female refugee participants, both Congolese and Iraqi participants mentioned the welcoming nature and sense of belonging they experienced in their community. This sense of belongingness was established upon their initial resettlement; *“it feels like it’s your home the first part that you arrived in”* (IJF2). In fact, when asked if they felt welcomed, all Iraqi refugee focus group participants indicated that they felt welcomed by their host community upon arrival to the United States, as if *“you’re part of America”* (IJF1).

When asked about the possibility or desire for future migration in the U.S., as they began to build their social network outside of their immediate city of residence, participants mentioned the ease to move to a place where they felt more belongingness and connected:

*“People in New York. People in Indiana* [have moved elsewhere outside of [City] and they feel more welcomed]*. Like some friends will meet them, but in Indiana they enjoy. If we tell them what we are facing, they are like just move”*(COF3).

Participants indicated that both *bridging* (host community) and *linking* (social service agencies) ties contributed to their sense of belongingness. Participants acknowledged the kindness (social and emotional support) they received from their local neighbors and community: *“For now we have to stay over here because people to be honest with you from here they’re really kind and nice”* (IJF4).

One Congolese participant tied their sense of ‘welcoming’ to the direct support (healthcare, employment) they were receiving:


*“You have come here, but you have not yet get any card. You only benefits is in Orlando. Its all America. They’re the same, but you be welcome in America. You have to wait until you get all those benefit. You’ve got the hospital. You work, and at least you have something for you to move”*
(COF3).

Although the sense of belongingness was more clearly articulated among Iraqi participants than Congolese participants (who expressed mixed feelings of belongingness), both focus groups acknowledged that welcoming took time. Successful outcomes and integration take time to adjust and develop social connections. Starting out, refugee women felt *“alone”* (IJF4) until relationships started to develop (*“they were talking and making jokes”* (IJF4); *“*[we started to] *talk and* [then I] *trust them”* (COF)) and feelings of belongingness within their host community began to shift *(“and we’re happy now”* (IJF4); *“step by step we were okay now”* (IJF4)).

Iraqi participants expressed an essential shift of feelings of belongingness when trust was developed. Towards the initial stages of resettlement, most Iraqi participants highlighted that trust was only found through religion (*“First she trusts God and then she trusts the government, then* [resettlement agency]*”* (IJF4)) and family (*Interviewer: “How much do you trust your husband to do their best? ”… “Alot.”* (IJF3); *“Because he’s* [husband] *the only one in here that I need to talk and trust him”* (IJF4)). After some time has been allotted to develop social connections, Iraqi participants were then able to develop mutual trust among *bridging* and *linking* networks, such as friends (*“Friend that you really trust”* (IJF4)), host community members, and resettlement agencies (*Interviewer: “Pick the person you trust the most.”…“Case worker*” (IJF2)).

### 3.2. Bonding Capital

**Church and religious social connections.** Research has shown that relationships established between refugees and the wider community differ by gender; for example, women may be more likely to make friends with other parent(s) at school, while men tend to develop their social network through work [45]. Despite the aforementioned isolation in the present study, several women have found social connectedness and support through church and religion. Church, one of the most frequently discussed social connection, provided both Congolese and Iraqi participants with social (*“She will pray to God. Talk to her kids”* (IJF4)) and emotional support, as well as a heightened sense of trust (*“If we didn’t trust the people from here or trusted God, we wouldn’t be here* (IJF4)) above most other social connections (*Interviewer:* “*Who do you trust the most?” … “Family first. Church is second*” (COF)). Even in an emergency, one participant said, “*Call 911 and pray to God*” (IJF3).

**Work social connections *(Congolese*).** Congolese participants developed bonding social capital not only through religious outlets, but also through work and among their female colleagues. Unlike previous gender research [45], and in alignment with others (i.e., [90]), work was also an important place to build and develop social connections for female refugees. However, despite work being a source for relationship building, the development of strong social connections still remained fairly slow: *“Most of us they will say that we have friends from work. That is where you meet new people, because they are working with them. I get to know you slowly by slowly making friends”* (COF).

Work provided women with an outlet to leave the physical space of their home and focus on non-household responsibilities, and an opportunity to cultivate friendships: *“It’s difficult* [to make friends] *because you don’t work. You are just staying at home”* (COF).

**Supportive female social connections (*Congolese*).** Fellow female refugee relationships provided an essential support for a variety of problems that Congolese participants faced. Such social connections provided unique assistance to problems; they provided *“hope”* (COF) and guidance (*“show me”* (COF)) in moving forward with the issues they were facing. This aligns with previous migration research [91] that showcased that refugee women tend to establish higher levels of *bonding* (i.e., homogenous genders) versus *bridging* social capital.

**Fellow refugee social connections (*Iraqi*).** Iraqi refugees expressed a shared understanding of the difficulty of the resettlement process. Especially after resettlement, Iraqi refugees had limited interactions with networks outside of their immediate family or ethnicity (*“Everyone is Kurdish”* (IJF3)). They also missed many aspects of their homeland that they had to leave behind, as well as parts of their home culture that was starkly different from their new host community’s culture. When they find others from their same culture or experience, this commonality provides an instant bond and underlying desire to support one another in the same or similar challenges. One way an Iraqi woman developed connection with her fellow refugee neighbor was by bringing him food: *“Because there is a neighbor right here and he’s a old guy but when he cook something we always give it to him, because he’s a refugee too, so we helped him”* (IJF4).

**Family (*Iraqi*).** In contrast to Congolese participants, Iraqi participants would often turn to their only social connections, which was often their family members (*“mom,” “father,” “husband,” “boyfriend,” “sister,” “brother,” or “child”* (IJF2, IJF3, IJF4)), or in some cases, close friendships. Family members became available and were immediately accessible for support because they were some of the only social connections that participants interacted with regularly, even if they were not physically nearby; *“when I talk to them* [my sister on FaceTime]*, I feel better”* (IJF4). When asked about various characteristics of their social connections, including qualities of importance, power, and safety, participants generally prioritized family due to their reciprocal trust (*“close in trust”* (IJF3)) developed among these relationships.


*“Does she have any close relationship here with people that are not Kurdish? “No. Just only she has... She just comes to our home and goes to my sister’s home, that’s it. Her mother-in-law and sister-in-law and that’s it. She doesn’t have anybody just only us”*
(IJF4).

### 3.3. Bridging Capital

**Language and culture.** Participants in both focus groups acknowledged that making friends is *“not really easy”* (COF), *“very difficult”* (COF), and *“hard”* (IJF2). For several women, developing bridging connections is a slow and burdensome process, due to differences in language and culture with their local host community. Language proved to be a significant barrier to developing *bridging* social capital, making connections with differing ethnicities or American host community members challenging. Integrating into their host community was *“harder... because another language”* (IJF3, IJF4).


*“One thing which I would say is making friends depends on someone. It depends on someone, how that person is, their personality, how they talk. Then what could be also to be challenge on the refugee women, most of them they don’t speak their language. How can you make a friend with someone? You can meet me but I want to become your friend. We don’t have the collaboration in language”*
(COF).


*“Let me say it in general. Whites are not like us Africans. For us African we are kind of friendly. I hope you understand that, but Whites they have that kind of something which is kind of privacy but for us as we grow, as our culture, I can come to your house. I go there waiting for the food. I eat. Something like that but that is not of the Whites. I have to come at the right time, if I was invited”*
(COF).

**American social connections (*Iraqi*).** In some instances, Iraqi participants suggested that the connections made with their American host community were just as essential as relationships developed among fellow refugees. These social connections provided essential trusting (*“Yes, I have a friend, she’s American and then I also have another friend he was an army that I trust him a lot”* (IJF4)) support for refugees post-resettlement. However, the nature of these ties varied in strength; while fellow Kurdish refugees provided emotional and social support, American connections were more transactional with direct, tangible assistance. Two participants described the transactional relationship through their encounter with their American friends at the airport. The interviewer asked, “Do you have any close relationships with people here that are not Kurdish?” The participants responded: “*They’re Americans. One man and woman... they came to pick us up from the airport*” (IJF3) and “*Some American... by the airport when they come and pick us [up]*” (IJF4). Host community members often helped refugee participants with navigating complex U.S. systems and helped them feel welcomed when they first arrived by picking them up at the airport. One participant acknowledged the limited strength of such ties with their host community, where support was provided under the conditions and convenience of the community member: *“A couple of friends if we need anything call them and when they’re not busy, they come here and help us. Yeah, the Americans”* (IJF4).

### 3.4. Linking Capital

*Linking* social capital [50] proved to be the most challenging for participants to establish and further build upon. Participants mentioned this difficulty, starting with the first day they resettled to the U.S. Access to resources or additional social connections for support was very challenging, due to a lack of cultural competence, as well as perceived apathy and lack of emotional connection among resettlement service providers. Congolese participants also expressed the obvious deficiency of social service resources, poor communication, and language barriers that further inhibited their development of *linking* social capital. Due to these obstacles, building and developing social capital is very challenging. Refugees are left to advocate for their needs, find their own solutions to everyday stressors (i.e., broken phone, internet issues), or navigate complex U.S. systems by themselves, due to the limited support from social service agencies or other community resources.

**Essential information and resources (*Iraqi*).** Similar to their *bridging* connections, Iraqi participants also established weak ties among their *linking* connections to gain essential information and resources. Although somewhat transactional, these networks (resettlement agencies) were critical for providing them with ‘the most information since coming to the U.S.’ (Interviewer question); *“...if I need to go somewhere they [resettlement agency] can come pick me and take me and find me some ways to do good things”* (IJF2). Although weak, these institutions also provided a direct link to essential resources to support refugee health and well-being, extending beyond the initial point of resettlement: *“Whenever I need them [resettlement agency], twice a month maybe... they gave us social security, DCF, Medicaids”* (IJF3).

**Lack of cultural competence (*Congolese*).** Starting with their first day of resettlement in their new host community, Congolese participants mentioned difficulty with building strong connections with *linking* institutions due to the organization’s lack of cultural competency and sensitivity.

Upon arrival into the U.S., refugees are required to repay (reimburse) their travel expenses on the interest-free loan that was used to cover the cost of their airline ticket provided for refugee resettlement to the U.S. [92]. Refugees are then allegedly supported by agencies that specialize in resettlement services. The U.S. State Department provides refugee furnishing, rent, food, and clothing for three months; nonprofit, charity, or faith-based community members provide all remaining support during this time.

Following this brief period of governmental support, assistance shifts to the Department of Health and Human Services and other nonprofit organizations that provide additional supportive services (i.e., medical assistance, language classes, employment preparedness) [91] through a public-private partnership model that draws from (generally faith-based) local community networks [93]. Although well-intentioned, one Congolese woman specifically remarked on the contractual feeling of this weak social tie, providing resources (i.e., food, furniture) with limited knowledge of cultural differences:


*“They take money. We are from Africa. We don’t know. We don’t eat food. Then, they come. What do we eat? If they... a bigger thing for the [church support] and then they cannot put in the house. On the first day you come, and you don’t know anything. They will start counting even those thing food. We got this. We bought this. Only your money. We don’t even wait for you to come because they don’t know the food you eat”*
(COF3).

**Perceived apathy (*Congolese*).** The common sentiment of caseworkers and social service providers *“not even caring”* (COF) was mentioned several times by Congolese women. Participants acknowledged that this perceived lack of genuine care and *“emotional connection”* (COF) stemmed from the service provider being overworked (“*She* [service provider] *was like, ’Oh that day I will be so busy. I have two jobs’*” (COF)), as well as having a lack of language commonality (“*If I can easily communicate with you, most likely I’m going to look after you. If I cannot, emotionally I’m not connecting, language is a barrier, you are a gone case*” (COF5)) and lack of desire to support or connect refugees with additional resources (*“it’s there, but they don’t want to give it to you”* (COF)).

**Poor communication (*Congolese*).** According to Congolese participants, alongside (and perhaps hand in hand with) culturally insensitive and deficient resources resides poor communication between social service providers and their refugee clients. Participants remarked that the most frequent point of contention, in terms of ineffective communication, was the lack of availability over the telephone. Many participants emphasized the lack of answering phone calls and responding to voicemails in a perceived timely manner. Through their elaborations and narratives, participants provided further elaboration as to the frustrations and expectations of what constitutes ‘timely’ and ‘effective’ communication.

According to participant experiences, social service provider or caseworker responses to client phone calls included: “*she was like, ‘I’m busy*” (COF3), “*sometimes they don’t answer the phone*” (COF3), or “[they] *take time to... answer*” (COF3). This ‘time’ to call back was quantified by two participants as “*after two or three days*” (COF6) and “*hours and hours... maybe after two days*” (COF1). Considering the lack of power and social connections held by refugee women, especially after she “*keep[s] on calling*” (COF3), or in urgent or emergency situations, these unmet expectations for timely communication may cause further vulnerability, feelings of distrust, or isolation.


*“One thing so far, why do they put their numbers in voicemail every time? Someone can have an emergency. You call them, maybe you have an emergency. I’m new here. I don’t want a policy like that”*
(COF1).

Several participants acknowledged that the wait time for support and communication response is lengthy, even when meeting for in-person care as well:


*“If you go there [“charities”], they will make you sitting there. You wait for long time then they attend you. They say go back. I already made a client in the internet so if they answer I will call you. It’s just like that. Then you come, you sit, wait for even one month”*
(COF).

Despite their deficient allocation of services and poor communication tendencies, participants still mentioned that these entities were (at some point in their resettlement process, to varying degrees) a resource where *“you have to go”* (COF). A sense of familiarity and necessity, despite its frustrations and weakness, was paired with this social *linking* tie.

**Language *(Congolese)*.** Similar to *bridging* social capital, language proved to be a significant barrier among Congolese participants in developing *linking social* capital as well. Language was a barrier to accessing appropriate help or resources from authoritative institutions of power, such as resettlement agencies, emergency services, healthcare facilities, and the police. One participant felt, *“If I can easily communicate with you, most likely I’m going to look after you. If I cannot, emotionally I’m not connecting, language is a barrier, you are a gone case”* (COF5). In another instance the misunderstanding from the language barrier involved a moment of emotional distress and further vulnerability by bringing in law enforcement:


*“The person who knocked called the police. They had a misunderstanding of the language. The police was asking, “What can I do?” Committed the kid to the hospital. The kid was hurt, the mother was crying, shaking. After a minute, and then someone came, “Those people they’re speaking like African language”*
(COF1).

**Lack of quality social service resources.** One of the most prevalent themes among all participants that emerged as a barrier for support and well-being was the deficient lack of quality social service resources. Congolese participants specifically raised several issues of retention and high turnover *(“they just sent me to another person”* (COF)), increased caseloads, and shortened times for availability (*“I’m calling because I know if I call her after 3:00 PM, she will be off”* (COF1)) for support.* “There are some agencies are overwhelmed. Case specialists or case managers are being given a lot of cases that, I would say, legally it should not be allowed”* (COF5).


*“We’re running through that a lot, but the biggest concern I am having is the case specialists having too many cases to handle and they are dropping the ball on someone. All the agencies in town, yes”*
(COF).


*“They have also called the food stamps. Then they know that she’s not working. The income of the two, its very low. In her age, she doesn’t know how to read. She doesn’t go to school. really the government, by the time they knew about her, about all the clothing she have. really is the government want just to leave her like that. Even the food stamp, it’s cut. Everything is cut”*
(COF3).

Both Congolese and Iraqi participants addressed the overall lack of quality *(“waste money for nothing which would help me”* (COF3), *“they are diving into mediocrity”* (COF5)) or insufficient services among agencies. When addressing daily issues or problems long term, both Congolese and Iraqi participants overall preferred *bonding* (family, other refugees) or *bridging* (friends, host community) social connections—or even no one at all (they would take care of the matter themselves)—in comparison to social service agencies. When asked what social connections were most helpful when they first arrived in the U.S., Iraqi focus group members mentioned: *“Yeah, but they* [friends] *help us more than* [refugee resettlement agency] *to be honest with you, yes”* (IJF4).

*“They* [friends] *help us more than* [refugee resettlement agency]. *They take us to interview like Walmart, when we need something we told them, but* [refugee resettlement agency] *just bring us over here and take care of the people that’s it, but they* [friends] *help us a lot”*(IJF4).

The lack of quality and quantity of social services resulted in refugee self-dependency and further perpetuated their isolation, creating additional stressors for participants that negatively impacted their well-being post-resettlement.

***Self-agency.*** Building social connections to access support, information, and resources is very challenging, and participants (especially Congolese women) conveyed the lack of facilitation or support from caseworkers and social service providers in this networking process. Therefore, many participants were forced to take matters into their own hands by advocating for and solving problems by themselves. Participants seemed to express a sense of feeling defeated by the lack of support, so they decided to move forward independently (“*I went there and I applied by myself*” (COF)). Such strategies have the potential to be problematic, considering the lack of social connections (and therefore limited access to resources, information, and support) and ongoing feelings of isolation that many had already previously expressed.


*“You have to face that by yourself. Then once those people [social service providers] start to return to you, they will just tell you they have no solution. Your benefit got finished. They will just give you number to try to call the government and claim the government’s got to help. That’s what they do”*
(COF3).

Many times, when Congolese participants reflected on which social connections they would reach out to for support when facing an issue, they instead discussed that they would solve the problem themselves. In response to whom they would speak or talk with about their problem, many responses resounded with a similar comment: *“Nobody because I don’t know any”* (COF).

To counter this barrier to *linking* social capital, refugee participants began self-advocating for their own needs, problem-solving the issues themselves, or sought out alternative networks for support. Some sentiment suggested that women already knew that the development of social networks would be a slow process; instead, they are self-advocating on behalf of future generations of female refugees:


*“It’s also better you talk, and sometime when you feel like you want to cry, you cry. We hope one day a chance. Others which are coming behind will get a chance. Only because we are thirsty. We won’t get a chance. No, that’s... for us. We are fighting for those ones who are behind. Maybe the children of the children only... those”*
(COF3).

***Isolation.*** Overall, both Congolese and Iraqi participants discussed the challenges with developing and maintaining social connections post-resettlement. The challenges of not feeling connected with their host community and a lack of linking support aided in negative outcomes. A lack of social connections caused *“suffering”* (COF)), loneliness, and social/emotional distress among participants; *“miss home, so we were coming from Arab to here, we don’t see anybody”* (IJF4). Isolation was also expressed through feelings of not receiving adequate support from social service agencies, resulting in further loneliness and negative health outcomes, especially among Congolese participants.

Lack of support in resources and information regarding housing, employment, transportation, childcare, food, and healthcare proved to be significant barriers to well-being. Additionally, considering the unique finances associated with refugee populations, such as the repayment to the federal government of the original resettlement flight to the U.S. (*“still we have to pay the ticket which brought us here”* (COF3)), these populations are starting with an unimaginable task with limited resources and support.

Many refugees emphasized the impact of employment–especially unemployment—on successful integration. Such *“low”* (COF1) or non-existent wages have significant impact on other factors, such as affordable housing.


*“You start working on the 17th. Then they tell we have let you go. You have to pay the rent. How can I pay rent when I start working on 17th? I have to wait that because when I’m working, they pay two weeks. I have to wait for those two week”*
(COF3).

The makeup of the participant’s family unit also played a significant role in how much employment (or lack thereof) became a significant stressor:


*“They give them the food stamps of the kid only thirty days for the whole month. They she’s working. They have to pay for housing. They have to pay for electricity. **They have to pay everything**”*
(COF3, emphasis added).

For women with children or without a husband, financial challenges became a significant inhibitor to well-being. Low income became a stark and difficult realization, considering the additional financial burden that is placed on mothers.


*“The challenge which I know that they are facing is her with a daughter and a son, the income is very low”*
(COF).


*“At least you know, “My child she’s on day care. It has worked peacefully. Let me go to work.” Other they’re failing to go to work because of the kids and then at the same time, if it does not work they cannot afford a house”*
(COF1).

For single refugee women, the lack of a dual-household income and domestic support also places additional compounding pressure to pay for all the household expenses:


*“The good thing, it would be better if the husband is around because the husband could be also working. At least that’s more income they could be sharing and they share the bills. Now only two people in the house are working and then the income is very low”*
(COF).

Several participants highlighted the high expense and burden of rent. Affordable housing cannot be obtained; what minimal affordable housing is available is not large enough or adequate/appropriate housing for a family unit.


*“One month you are telling the landlord to wait at least, to hold on. You don’t have the money, but once you get your payment you pay that, but in America, it’s all like that. Once you given your purse on day, they challenge you for that. That one day. You have to pay a hundred and something”*
(COF3).

Considering that a majority of the study’s participants are renting their housing, this leaves a heightened possibility for continual migration and a lack of permanency in their dwelling space, which further perpetuates risks of vulnerability.


*“They have an issue with like whether they’re working. Now where they are staying, the owner of the house want to sell the house”*
(COF3).

For several refugee participants, due to unattainable housing costs, alongside compounding taxes and credit interest, the security of housing was prioritized over food for the family. However, one participant mentioned that, with the limited financial security of employment (despite the low income mentioned previously), there is an opportunity to pay for *“at least my housing and get my food”* (COF3).


*“When she failed to pay the rent, she has to go and take a credit. Then you have to take 500. You have to return that 555 dollar. Then after paying that... when you are getting 1200. Thats not just that passed into that check. Then you pay the tax. You are paying the house maybe 800 or 900. You have nothing put on food”*
(COF3).

Such stressors not only have serious implications for mental and emotional well-being, but also physical well-being. It was aforementioned that the prioritization of food (nutritious or otherwise) was low, in comparison to other living essentials (such as housing). Additionally, medical bills and affordable healthcare were also indicated as financial challenges and stressors, which proved to be a significant burden. With a lack of employment, many participants lacked any healthcare benefits; *“She doesn’t work. On the top of that she doesn’t have Medicaid”* (COF).


*“If she throws sick or go to the hospital, the bills do come very much. She doesn’t work. How is she going to pay them?”*
(COF3).

## 4. Discussion and Conclusions

Drawing from the results of this study, the researchers conclude with a discussion on future research directions and tangible recommendations for policy considerations, integration process revision, and infrastructure development for resettlement advocates and practitioners.

### 4.1. Moving beyond Homogenous Identities

In alignment with previous research [26], the diverse, complex history, and unique concerns [24] of a female refugee’s ethnicity must be taken into account in relation to health outcomes and well-being. Considering the differences observed between the focus groups, refugee programming and policy must take refugee ethnicity (as well as other intersecting identities) into account. Conjoining ‘all refugee women’ into one distinct category or group is a disservice to the oppressed identities and advantageous strengths that each individual holds. For example, in the present study, Iraqi participants highlighted that extended family, friends, and *bridging* social ties were essential to support their well-being significantly more than their fellow Congolese women. Researchers, policymakers and social service providers must consider the intersections of gender and race, ethnicity, and national origin [66]. An integrated, multi-system approach is needed to improve female refugee well-being and health outcomes [24]. For policymakers specifically, this includes combatting gender neutral terms in migration policy, which ignores power dynamics and socio-economic and -cultural systems that create a differentiated gendered experience [9]. Echoing previous calls for social service reforms to build knowledge and skillsets for navigating U.S. society [5], the authors also encourage resettlement agencies to include the historical contexts of race, ethnicity, national origins, and gender as they relate to the United States as part of the cultural orientation training for refugees post-resettlement.

### 4.2. Belongingness and Trust

While Iraqi women still experienced loneliness and isolation post-resettlement, results suggest that they emphasized their level of connection and sense of welcome with their host community more than the Congolese women. Due to this increase in vocalized social connectedness, the researchers consider the following questions: Are Iraqi women better connected in terms of *bridging* social capital? Do they experience a greater sense of belongingness accordingly?

Results also suggest that this heightened level of belongingness stems from a shift (*“it’s just different”* (IJF)) in trust among their social connections. Previous trust literature among migrant populations defines ‘trust’ as a community’s capacity to generate reciprocity among residents [94]; this reciprocity suggests that there is an expectation that relationships developed within the community will be available when needed [48]. The results of this study align with previous scholarship, which concludes that, in order for social networks to develop into social capital for the individual, the relationship must be built on reciprocal trust [95]. When considering the complex nature of refugee resettlement, trust becomes more of an abstract concept. Refugees need to place trust not only in their social connections, but also in the larger “humanitarian commitment” (p. 338) of social services and aid within their host country [96].

Iraqi participants in the present study highlighted that trust was found first through religion (*“God”* (IJF)), then family, and later (after some time was allotted to develop social connections) among *bridging* and *linking* social ties. Trust is a significant factor especially when developing *bridging* social capital; this capital stimulates cooperation and familiarity, leading to knowledge-based, interpersonal trust and an increased generalizable trust [50].

The authors found it interesting that, although religion provided a foundation for the development of trust, there was a lack of discussion among participants regarding religion as a facilitator of distrust among non-Muslim community members or possible discrimination, as discussed in previous research highlighting gender-based, anti-Muslim bias and prejudice in the U.S. (i.e., [3,30,33,34,35,36,97]). Future research on how this spiritual connection impacts other social networks or contributes to other aspects of refugee well-being or belonging should be explored. For example, although not well studied, how might anti-Muslim discrimination influence economic outcomes [30] or financial stressors discussed in this study?

Stolle et al. [98] explored the impact of ethnic diversity in Canada on social unity, specifically considering its relationship with trust. They found that individuals who consistently engage with their neighbors are less influenced by the ethnic characteristics of their surroundings than people who lack consistent social interactions. Therefore, social networks could mediate negative effects of diverse populations on trust; ongoing social interactions among “diverse others” [98] make ethnic differences less threatening. Drawing from these findings, in the present study, results suggest that ongoing social connections with ‘diverse others’ (refugees of other ethnicities, American host community members, resettlement agency personnel) could have shifted perceptions of Iraqi female refugees to become more trusting, despite their differences.

Conversely, several consequences arise from distrust of social connections. Due to their limited contact or exposure, the distrust that many refugees hold towards both service providers and their host communities [99] further perpetuates a state of vulnerability and isolation. The absence of trust as an emerging theme among Congolese participants in the present study is concerning, possibly contributing to or even bolstering negative outcomes yet to be explored.

### 4.3. Social Capital

Building on previous research [47], the results of this study also confirm that refugee social connectedness has a gendered dimension regarding social capital resources. Bonding, bridging, and linking social capital proved to be essential to refugee well-being, providing various degrees, ways of developing, and types of support.

**Social bonding.** To support women’s successful integration, trusting their social connection of kinship was more important for Iraqi than Congolese participants. The present study contributes to previous research [47,92] that suggests that higher levels of *bonding* social capital (i.e., family, close friendships) exist among female refugee networks than *bridging* social capital. Such strong kinship ties have previously been shown to improve the quality of life for refugees, ironically for men more so than women [100]. However, research has also suggested that an increase in *bonding* social capital has the potential to make it harder for individuals to trust their surrounding diverse community [101]. Did Congolese participants experience heightened levels of distrust towards their host community? Although not directly seen in the present study’s findings, the authors suggest that further migrant research considers the gendered nature of trust among highly bonded refugee communities.

Despite the isolation many participants experienced, the results of the present study concluded that the church was an important network in the development of social *bonding*, contributing to social and emotional support, while spirituality (*“God”)* was rooted in significant trust. Findings support previous research [30,42,47,54], affirming the contribution of religious institutions to essential socioemotional support for female refugees. Similarly, the trust of spirituality vocalized among participants in the present study could also contribute to female refugee resilience and coping, as shown in previous research [4].

The authors note the lack of religious dependency as expected in the Congolese female focus group. During the time of this study, as previously mentioned, almost all (96%) Congolese refugees were pronounced Christians, finding churches to be a tremendous resource for both fiscal and emotional support upon their arrival, similar to their lives in turmoil back in the D.R.C. [73]. Many U.S.-resettled Congolese turn to religious faith for healing and support as an important role in personal and community life [83]. Perhaps these results point to a possible shift in spiritual dependency as a significant source of *bonding* capital or coping mechanism among female Congolese refugees.

**Social bridging.** In alignment with previous research [102], the workplace proved to be a source for Congolese participants to build and develop social connections. These results contradict previous gender research that has suggested that men (more so than women) tend to develop networks through work, while women may be more likely to build relationships with other parents at school [47]. These differing results could be due to the demographic make-up of Congolese women in the present study, considering that most participants were never married and had little to no children living in their homes, encouraging them to spend more time at work (to address their single household income) and limited interactions with parents at school.

Regardless, considering the importance of the workplace as a platform for social connectedness, the authors suggest that future research should consider the importance of employment not only to combat financial stressors emphasized in this present study, but also as an opportunity to further develop *bridging* social capital. Economic integration research among U.S.-resettled refugees tends to focus on how fast refugee groups can find employment [5], as well as how much they rely on support services beyond the initial 90-day federal resettlement support [103], versus the opportunity to develop social connections through the workplace.

Previous studies have highlighted the low levels of employment among highly educated refugee women and the lack of economic niches for those less educated, which significantly impacts their lower earnings [33]. Employment has also been shown to vary depending on gender (women experience lower employment rates than their male counterparts [16]), as well as the length of time an individual has resided in the U.S. [104] and citizenship status [105]. While female refugees have lower levels of employment within their first year post-resettlement (especially women who have had children more frequently and earlier [105]), male and females experience have shown to hold similar employment effects in later years [106]. However, it is important to note that other research has yielded mixed results and employment differences among gender; for example, one study concluded that U.S.-resettled refugee women are actually employed at a higher rate than their male equivalent [107].

Despite these mixed results, an overwhelming about of literature points to the significant benefits of employment for refugee women, not only for themselves (i.e., greater autonomy, opportunities, social support, and combatting isolation) [108], but also for the economic vitality of their host country [105] and community. There are many challenges that face employment research among migrant populations, such as the limited refugee labor market data and scarce disaggregated refugee data by gender [109]. Additionally, challenges arise with data collection and analysis, considering the interdependent barriers and political and social issues that vary across contexts and impact employment data [109]. Although the authors acknowledge these many obstacles, they still promote the importance of future research on exploring the gendered differences of employment as forms of both *bridging* and *linking* social capital.

While the multiple identities of female refugees must be considered to better analyze the development of *bridging* social capital (not only among work colleagues), the results of the study also suggest that relationships established between refugees and the wider community are gendered and take time to develop. The lengthy and difficult process of making friends is due to differences in language and culture with their fellow refugees of different ethnicities and languages, as well as their local host community. Female refugees disproportionally lack *linking* social capital such as career readiness or education courses [49] and are less likely to have access to social services that provide English language classes [51] in comparison to their male counterparts. This directly impacts their ability (or inability) to make social connections with individuals in their host community.

The difficulty expressed by participants in developing *bridging* networks could also be due to tensions in identity development. For example, a case study approach was used to identify how female refugees construct social networks and how these networks aid or restrict identity development in a new place [19]. Similar to the results of the present study, many of the women held weak (or non-existent) ties with members of their host community. Missing that strong *bridging* social connection restricts network connectivity [19]; *bridging* social capital is fairly limited with many refugees. Relationship ties formed during the rebuilding process impact identity redevelopment, as female refugees attempt to balance the past self with the future self and mourn the loss of their old self. The authors suggest that future research explores how the rebuilding of female identities is shaped by the resettlement process among varying demographics. For example, in the present study, Iraqi participants suggested that the connections made with their American host community were just as essential as relationships developed among fellow refugees, although the nature of these ties varied in strength. Did the strength and quantity of *bridging* ties among Iraqi women alter their rebuilding of identity post-resettlement? Was this process different for Congolese women, who possibly had less *bridging* social capital? Future migration research should attempt to address these questions through an intersectional lens.

**Social linking.** Congolese and Iraqi participants held varying perspectives on how social service institutions provide support. While Iraqi participants capitalized on these transactional weak ties to gain essential information and resources to support their well-being, Congolese participants are caught in the tension of acknowledging their necessity for support and the frustration of the lack of cultural competence, perceived apathy, lack of emotional connection, and poor communication among these institutions. This discontented feeling is concerning, when considering the significant stressors emphasized by Congolese participants; increased distrust and discouragement surrounding a lack of support leads to further distress. The authors consider the unique dynamics of these findings, when considering other previous research, such as the Refugee Well-being Project [99]. Data from a large, randomized control study called the Refugee Well-Being Project used a mixed methodology to answer how social ties impact the refugee resettlement process in the U.S. among refugees from Afghanistan, Iraq, Syria, and the Great Lakes region of Africa. Both strands showed that stronger ties with service providers generally created greater distress [99]. A reason for this dynamic comes from the transactional relationship that service providers tend to have with refugees, where they are called in a time of crisis or great need; this contrasts to the transformative relationships established with non-service provider support found in bonding and bridging ties.

Both focus groups agreed that *linking* social capital was the most challenging to establish, where a deficient allocation of quality social service resources left refugees to advocate for their own needs, find their own solutions to everyday stressors (i.e., broken phone, internet issues), or navigate complex U.S. systems by themselves due to the limited support from social service agencies or other community resources. Resources obtained through *linking* social capital have significant implications not only for the development of *bonding* and *bridging* social capital, but are also essential for them to thrive in their integration process [47].

As aforementioned, the lack of quality and quantity of social services resulted in refugee self-dependency and further perpetuated their isolation, creating additional stressors for participants that negatively impacted their well-being post-resettlement. Considering that many obstacles that contributed to refugee isolation and loneliness were caused by financial stressors, the authors suggest that refugee advocates and policymakers shift an intentional focus on cultural orientation courses specifically aimed at financial literacy with an understanding of the U.S. credit system.

Additionally, prevention strategies for social service providers are essential in addressing the mental health symptoms resulting from the entire resettlement process, including pre-migration trauma, the difficulties and challenges with initial integration [42,43,110], and significant hardships and trauma that post-resettlement [11]. Such prevention strategies have the potential to impact other areas of refugee well-being, beyond the direct mental health benefits. For example, the mental and physical health among refugee populations has been associated with their educational attainment and English language ability [31], which were both found to be potential obstacles to further social capital development in the present study. One way to address these mental health issues (especially heightened levels of loneliness and isolation) is by partnering with ethnic-based community organizations (who have established trust) to deliver psychoeducational materials and workshops [26]. Drawing from previous suggestions (i.e., [24]), social service providers can work with community leaders to foster dialogue, education, and collaborations among local organizations to bolster community social networks. Such community engagement strategies have been shown to improve mental health symptoms among new arrivals to their host community [111]. Despite the increase of community social support, it’s important to note the aforementioned challenges female refugees face when navigating the U.S. healthcare system [22,23]. Therefore, there is a further need to address language barriers and improve overall provider-patient communication [24]. Considering the variations in strength among female refugee social network ties, and their corresponding impacts on refugee access to support, information, and resources, future research should be conducted to better understand the types of networks and their degrees of support. Future questions include: If participants had more social connections, would their health and (in turn) well-being be positively impacted? How does the strength of their social connectedness influence everyday stressors? What about in emergency or crisis situations? How do social connections influence levels of loneliness and isolation? In alignment with previous research, the findings of the present study suggest that an increase in social capital could provide opportunities to enhance refugee health and well-being; however, refugees listed a multitude of barriers to forming social connections that ultimately increased their loneliness and isolation, contributing to further vulnerabilities and lack of resources. While each person will experience barriers differently based on various cultural and demographic identities (including gender), each focus group did face a multitude of barriers. What barriers could be combatted with strengthened bonding, bridging, and/or linking ties? Further research is needed to identify, address, and increase policy interventions to curb such barriers.

### 4.4. The Role of the Researcher

Considering the CFP lens used to explore refugee social connectedness in the present study, the authors note the unequal distribution of power held by the researcher. As aforementioned, the researchers acknowledged reflexivity, considering the power of the research findings on the population and the limitations of the authors’ perspectives [78]. However, it is critical to note that both focus groups were initially somewhat distrustful towards the research process, suggesting that the researcher had affiliations with governmental entities, even questioning the altruism of her intentions. The researcher clarified her role as a representative of a university and provided culturally appropriate compensation to show appreciation for their time and investment in the study. A commonality between the researcher and participants is their identification as cis-gender women. While we all experience gender differently, some subliminal shared identities could create trust or credibility. Regardless, the role of the researcher as a position of power among female migrants (among all vulnerable populations) must be considered in the entire research process: its design, implementation (data collection and analysis), and distribution of the results.

### 4.5. Limitations

The small sample of the present study limited our analysis capabilities; due to the sampling approach, results cannot be generalized to other refugee populations with similar identities. Considering that both focus groups were female only, there was also a lack of a comparison group of male refugees; therefore, the authors were unable to compare the unique experiences of resettlement to that of their male counterparts [4]. Additionally, since the focus group process and guide were translated from English to the participant’s native language, then back once again for the researcher, there is the potential that the questions were not fully understood or culturally resonated with participants, leading to further misunderstandings, o even possibly a loss of nuance to the posed questions. However, the results of this exploratory study can still lend powerful insights to the impact of female refugee social connectedness and levels of belongingness and well-being.

Lastly, the authors only explored cis-gendered, heterosexual refugee women. Among migration research, there is a significant lack of attention paid to the health (including sexual/reproductive rights, risks, and needs) and well-being of queer immigrant women [9,19]. Therefore, the authors call for future migration research that takes an intersectional approach to highlight diverse gender identities that might otherwise be overlooked.

## 5. Conclusions

U.S.-resettled female refugees face unique challenges and barriers compared to their male counterparts. Previous literature has primarily focused on the obstacles facing refugees during the resettlement process without discussing the differing gendered experiences. This paper adds to the recent emerging research and conversation revolving around how different genders experience the resettlement process; this includes all levels of engagement with federal, state, and local systems, as well as informal systems such as neighbors, friends, work colleagues, and communities. While support is generally needed by both formal and informal systems, each one plays a distinct role in the refugee’s unique resettlement experience, with social service providers providing more transactional interactions. The informal networks of neighbors, friends, and community members provided critical social and emotional support. These transformational relationships strengthened *bonding* and *bridging* social capital, improving overall well-being. The strong isolation experienced among refugee women highlights the need for changes in resettlement policy and supportive programs to enhance gender equity among U.S.-resettled refugees.

## Figures and Tables

**Table 1 ijerph-20-05324-t001:** Female-only focus group demographics comparison.

Demographics	Congolese	Iraqi
City of Residence	Orlando (*n =* 6)	Jacksonville (*n =* 6)
Primary Language	French (*n =* 5)	Kurdish (*n =* 5)
	Swahili (*n =* 1)	Arabic (*n =* 1)
Secondary Language	English (*n =* 2)	English (*n =* 3)
	Lingala (*n =* 1)	
	Kikongo (*n =* 1)	
	None (*n =* 2)	None (*n =* 3)
Average age	*M* = 45.5 (SD = 14.511)	*M =* 32 (SD = 7.16)
Average English comprehension (read and understand)	*M =* 2.5 (Slightly well–Well)	*M =* 2.17 (Slightly well)
	SD = 1.00	SD = 1.34
Marital status	Married (*n = 0*)	Married (*n =* 3)
	Separated (*n =* 1)	Separated (*n =* 1)
	Widowed (*n =* 1)	Widowed (*n =* 1)
	Never Married (*n =* 4)	Never Married (*n =* 1)
Employment	Not employed (*n =* 4)	Not employed (*n =* 3)
	Employed (*n =* 2)	Employed (*n =* 3)
Education	*M =* 8.8 years (SD = 3.56)	*M =* 3.4 years (SD = 6.29)
Length (range) of time living in City	2 months–8 years	6 months–6 years
Rent vs. Own housing	Rent (*n =* 6)	Rent (*n =* 4)
	Own (*n =* 0)	Own (*n =* 2)
Number of individuals in household (average)	*M =* 2.83 (SD = 0.90)	*M =* 4.17 (SD = 0.90)
Total number of children (under 18 years old) in household	2 children (*n =* 0)	2 children (*n =* 1)
	1 child (*n =* 2)	1 child (*n =* 0)
	0 children (*n =* 4)	0 children (*n =* 5)

**Table 2 ijerph-20-05324-t002:** Themes and codes.

Belongingness and Trust					
Bonding Capital		Bridging Capital		Linking Capital	
Congolese and Iraqi		Congolese and Iraqi		Congolese and Iraqi	
Church and religious social connections		Language and culture		Lack of quality social services Isolation	
**Congolese**	**Iraqi**	**Congolese**	**Iraqi**	**Congolese**	**Iraqi**
Work social connections Supportive female social connections	Fellow refugee social connections Family		American Social Connections	Lack of cultural competence Perceived apathy Poor communication Language	Essential information and resources

## Data Availability

The data presented in this study are available on request from the corresponding author. The data are not publicly available due to IRB restrictions of confidentiality.

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
