# Peer review of "“Step by Step We Were Okay Now”: An Exploration of the Impact of Social Connectedness on the Well-Being of Congolese and Iraqi Refugee Women Resettled in the United States"

_ijerph, 2023, doi:10.3390/ijerph20075324_

Round 1

Reviewer 1 Report

Thank you for your work with Iraqui and Congolese refugee women- it is important to disseminate your findings regarding these vulnerable populations in order to inform the work of others.

In general, I suggest you update your  references, work on making your sentences more clear, and concise, and  focus on recommendations for practice based on the the qualitative results of the focus groups

You  use many outdated references (over 65, some as far back as 1989); some of these may be relevant as seminal works, but most are not. Some references used are not relevant to the populations studied (eg references to Cambodians, Yugoslavians, Somalians, Burmese, lesbians)) and are misleading in relation to the focus of your work; experiences and adaptation/acculturation in various cultural/ethnic population groups can be vastly different, as you discuss in your paper.

Following are a few more specific recommendations:

Line:

24- Use updated stats to illustrate and define your statement re "displaced people"- there is a difference between refugees, internally displaced , and those displaced outside of countries of origin.

51, 56- unclear sentences- rework

62- provide an example

63/64 appear contradictory- rework

68-69- rewrite

80- "Integration process"- this may be an outdated term- - do you mean acculturation process, or is this a different concept you are describing?

92- are you referring rather to "resilience"?

134- unclear sentence

158- does not seem to fit here

280- You mention in the text that not all participants completed the surveys, yet you report an n of 12 - ??- please clarify.

286- unclear sentence meaning

325 run-on sentence

Other points:

Table 1 difficult to read and compare

Study design section - consider making this more concise

Reviewer 2 Report

Thank you for the opportunity to review this manuscript. The authors use a qualitative approach to explore the perceptions of women of refugee background from Iraq and Congo post resettlement on belongingness and social connectedness throughout their integration into US communities. In response to this purpose or aim, belongingness and trust, bonding capital, bridging capital and linking capital as themes which were then explored overall and with explanation of the differences between Iraqi and Congolese women.

There is much to like about this manuscript, including the thoughtful situating of the study findings within the broader research in this area and an engaging writing style. That said, there are also parts of the manuscript that could be substantially improved: (1) the organisation and length of the introduction; (2) the detail provided on the methodology; and (3) the presentation of the findings.

(1) The organisation and length of the introduction

The authors provide a thorough and interesting literature review, however this may be improved by reducing the length of the introduction. A large amount of detail is provided on the gendered nature of resettlement experiences, social connectedness, belonging and social capital in the refugee context; however, the reader loses sight of the study focus.  Similarly, the Critical Feminist Approach section is very informative; however, the level of detail distracts the reader from the purpose of this study. These sections would be improved by being more concise.

Additionally, when reading the Introduction, the reader is left wondering why women from Congo and Iraq have been chosen as the population of interest. Please consider including this information in the Introduction rather than Materials and Methods. It does remain unclear why these groups were chosen - are Iraq and Congo the most common country of origin for refugee background populations in Florida? Are they typically under-researched? What was the imperative for researching with these groups of women in particular?

(2) Detail provided on the methodology

While the choice of focus groups for data collection is well-justified, insufficient detail is reported on the remainder of the study methodology. Please address the following points:

- What was the thematic analysis approach? Was a methodology followed such as reflexive thematic analysis from Braun and Clark?

- There were only two focus groups, however three options for managing language needs were reported (English only, partially in English or entirely translated). Please clarify this.

- A representative leader is reported to have co-facilitated the focus groups. Please clarify what a representative leader is, what experience or qualifications they had re interpreting and how they (or their relationships with the other women) may have influenced the power dynamics within the focus groups.

- How was language and translation managed? Were the surveys administered in English or the participants' first language? Were they verbal or written? Were all of the focus group data transcribed or just the English data? What language were the focus group data analysed in? This transparency will help other researchers consider how to manage multilingual data collection and analysis too.

- Was any member checking conducted or were the findings corroborated by participants/stakeholders in any way?

(3) Presentation of the findings

The findings were engaging and insightful, thank you. With respect to the focus group findings, clarity would be improved with a model or diagram showing how the concepts of belongingness and trust, social capital etc interrelate for each group of women, or overall. A number of the quotes were also repeated multiple times. Please avoid this. If possible, please provide an ID or pseudonym with each quote to show that a range of participants (as opposed to a small number) are being quoted. Reporting when a quote has been interpreted/translated would also improve transparency of how language and translation issues were approached. 

With respect to the survey findings, the presentation of the table is somewhat unclear. For example, it looks like the French-speaking women are the only women who also speak English. Additionally, if reporting means, please also report standard deviation. It is unclear if "average number of individuals in a household" refers to a mean or median. 

Other areas for revision include:

- please report the aim of the study in the abstract; and

- focus groups are reported as having 6 participants each and later in the manuscript, as having 5-6 participants (please clarify).

This feedback is intended to be constructive. This manuscript is beautifully written, engaging reading and very insightful. I look forward to reviewing the revisions.
